# OpenReview forum: "Chain-of-Symbol Prompting for Spatial Relationships in Large Language Models"
_ICLR.cc/2024/Conference — Submitted to ICLR 2024_

### Official Review · Reviewer_LjHX · 2023-10-30

**Soundness:** 3 good
**Presentation:** 4 excellent
**Contribution:** 3 good
**Rating:** 6
**Confidence:** 4

**Summary:**

This paper presents a novel prompting paradigm called Chain-of-Symbol (CoS) Prompting that represents the spatial relations in the symbol chain. The authors claim that the proposed CoS outperforms CoT in all existing spatial QA benchmarks, with fewer tokens compared to CoT.

**Strengths:**

1. The proposed CoS prompting and findings are very interesting!
2. The paper is overall well-organized and easy to read.
3. The experimental results are strong.

**Weaknesses:**

1. The step (ii) in CoS prompting is not clear enough. How do authors correct the zero-shot CoT’s output, is it manually checking or prompting LLMs to do so?
2. There is an overlap between “Natural Language Navigation” and the text in the box in Figure 2.
3. I am curious about the generalization ability of the proposed CoS prompting. I would encourage authors to report performance in more general domains rather than just spatial question answering, such as some open-domain QA tasks.
4. This paper presents very interesting findings, I would encourage authors to discuss more about its mechanisms and give some insights.
5. Though authors list some advantages of the proposed CoS, at the same time, CoS is more difficult to track LLMs reasoning path. I would encourage authors to discuss some limitations of this work.
6. The CoS prompting exemplars are generated with zero-shot CoT. I would encourage authors to discuss the motivation of selecting 0-shot CoT and its implications on CoS performance.

**Questions:**

N/A

---

> ### Author Response · Authors · 2023-11-15
>
> Thanks for your valuable advice.
>
> For weakness:
>
> 1. We manually correct zero-shot CoT's output, which described in Appendix B.2 .
>
> 2. Thanks for your kindly mention! We will adjust it in our paper.
>
> 3. In addition to spatial tasks, we add experiments on the temporal sequence QA tasks from the BBH benchmark which is shown in the third table of part 1 in general response for all reviewers.
>
> 4. Thank you for your encouragement. One intuition we think might be useful is that CoS uses symbols to replace part of natural languages, hence reducing noises brought from redundant natural languages. Neural network might only needs to know the relationship between entity A and other entities to conduct the complex reasoning or planning task. We postulate that removing irrelevant information contained in the natural language can help model to focus on the real task. And using symbols might can reduce potential context information gained from pre-training stage compared with natural languages.
>
> 5. We will add more limitation about this aspect in our paper. Although our method uses less natural languages compared with CoT, we actually contains some information of input in CoS ( e.g. objects / entities ), which is easy to trace them back to input descriptions, which can then track the reasoning path.
>
> 6. Zero-shot CoT is the most straightforward clue that one could have to use CoS in their own application. While other format can be useful, the search space is huge and it can be spend more time in writing the demonstration fully manually. Also, we tried in our preliminary experiments on human-written CoS and CoT. We did not have promising performance on both CoS and CoT when human writting, and we postulate that LLMs adapts better to the zero-shot CoT format produced by themselves.

---

### Official Review · Reviewer_c5jB · 2023-10-31

**Soundness:** 3 good
**Presentation:** 3 good
**Contribution:** 3 good
**Rating:** 6
**Confidence:** 3

**Summary:**

This paper proposes a new prompting approach, called chain-of-symbol. The idea is simple, instead of providing a language description about spatial relation of objects, this paper proposes to use symbols to represent the object relationship. By doing this, this method improves the off-the-shelf LLMs capability to reason about the spatial relationship. Five tasks are evaluated. The first task is called natural language spatial planning, in which the LLM is asked to move the object so that this location of objects will be in the desirable location. The second task is brick world, which is similar to the first task. And the third task is NLVR-based manipulation, in which the agent again needs to rearrange the object based on the textual instruction. And the fourth task is natura language navigation, in which the agent needs to navigate to different landmarks given the instructions. Finally, the spatial QA is a traditional benchmark for testing reasoning capability of LLMs. Table 1 and Table 2 show that CoS significantly improves the performance over CoT. And from Figure 3, the performance does not affect by the symbols that are used, which leads to that this method is robust to the symbols chosen. Overall, the paper demonstrates a nice result to verify the idea.

**Strengths:**

- The paper is easy to understand. All the sections are easy to read
- The proposed method is very simple, but the results show a significant improvement over CoT.

**Weaknesses:**

- This is another prompting paper in the era of LLMs, and there is no new algorithm, but a trick (finding) to make the LLMs more robust.

**Questions:**

- How about other tasks such as problems concerning the "sequential relation"? For instance, Alice was doing cooking this morning, and in the afternoon, Alice was working, and during the night, Alice was relaxing with the family. Question: what did Alice do between 12:00 PM to 6:00 PM? something like this type of question.
- What is the fundamental explanation that this method works well?

---

> ### Author Response · Authors · 2023-11-15
>
> Thanks for your valuable advice.
>
> For weakness:
>
> Although we don't propose new "algorithm", we provide insights to use symbols to replace natural languages in demonstration / prompting, which we think would be a promising direction to further explore for future works, for using symbols can wider the representation space in prompting, and reducing potential noises brought from natural languages. What's more, in our additional experiments about open-source LLMs, we find the scaling curve of CoS is steeper than CoT’s and the emergent ability of LLMs for understanding CoS (symbols) when size increases, which is very interesting. We also add different tasks of scaling curve in the latest uploaded version of the paper in figure 4, shows the general trend across tasks.
>
> For question 1:
>
> In addition to spatial tasks, we add experiments on the temporal sequence tasks from the BBH benchmark, which is very similar to the situation you mentioned, and results is reported in the response for all reviewers. The prompting demonstration and examples are added in latest uploaded version of our paper in the table 16.
>
> For question 2:
>
> We use CoS for two reasons:
>
> Not only using natural languages but also using symbols can make the representation space wider in prompting.
>
> CoS uses symbols to replace parts of natural languages, hence reducing noises brought from redundant natural languages. One assumption is that words of natural languages might have more context information during pre-training, which might bring potential additional information, but using symbols can avoid these, by connecting objects without potential additional information, and make prompting more clear and structural.

---

### Official Review · Reviewer_tveC · 2023-11-03

**Soundness:** 3 good
**Presentation:** 3 good
**Contribution:** 2 fair
**Rating:** 6
**Confidence:** 3

**Summary:**

This paper presents an exploration into the capabilities of large language models (LLMs) to understand and process spatial relationships, a relatively unexplored domain for such models. They propose a new method termed Chain-of-Symbol (CoS) prompting. This approach aims to improve the spatial reasoning capabilities of LLMs. They observe notable performance gains in tasks that involve spatial planning and understanding. For instance, in the Brick World task, which involves a series of steps to achieve a final goal, CoS considerably outperformed traditional CoT. The findings could have broad implications for tasks that reuiqre advanced spatial understanding.

**Strengths:**

1. The paper presents a novel method, CoS, which addresses the limitations of existing techniques in spatial understanding.
2. Multiple spatial tasks such as Brick World, Natural Language Navigation, and NLVR-based Manipulation were used to validate the efficacy of the proposed method.
3. CoS consistently outperformed traditional CoT, showcasing its potential as a superior method for spatial reasoning tasks. In addition, a thorough analysis of the results, considering different configurations are also provided.

**Weaknesses:**

1. The heavy reliance on symbolic representations might limit the model's flexibility in real-world scenarios where such symbols might not be explicitly available. There also lack a clear definition or universal guideline to convert tasks into condensed symbols.
2. The process of converting spatial tasks into symbolic representations could introduce additional complexity and computational overhead. Besides, it require annotations, which seems more difficult to obtain compared with natural language based chain-of-thought or program-based program-of-thought.
3. The open-source LLM remain under-explored, only ChatGPT series are tested in the experiments. It could have been more solid if the same trend can be observed in other LLMs. Besides, the design of prompt may also influence the results a lot, perhaps a through comparison on the robustness of the prompt should also be included, so that the CoT performance is really revealed in terms of capturing spatial relationship.

**Questions:**

1. How scalable is the CoS method, especially for larger and more complex spatial environments? Is there an automatic way to converse into chain of symbols, or does it require manual rules and symbols design for each task and even each in-context sample?
2. How does CoS perform when compared to other potential solutions or methods that might address spatial reasoning in LLMs? Such as program-based CoT that involve symbolic reasoning?
3. How is the computation required to further convert to chain of symbols? And how about the tasks that do not have manually annotated symbols, how to extend your CoS method to a broader range of tasks?

---

> ### Author Response · Authors · 2023-11-15
>
> Thanks for your valuable reply and advice, we add open-source experiments in the response for all reviewers.
>
> **For weakness**:
>
> Point 1:
>
> We will add a clear definition of universal guideline to convert tasks into condensed symbols in Section 3 of the paper.
>
> We first define our task in a more general perspective:
>
> We can formulate the tasks which our method is suitable as a set of tasks which has sequential and limited number of relationships, and the relationships can be spatial, and also can be other kinds such as temporal (which we extend as new experiments in the response for all reviewers).
>
> Then we define the procedure of our conversion from natural lanugages of CoT to our CoS:
>
> For converting natural languages into symbols, we can first use a recognizer to recognize objects and entities in the natural language descriptions, e.g. using human, NER system or even rule-based codes to automatically recognise objects in the sequential relationships described by natural languages, and then connect them with each other only using **random symbols** ( e.g. '/', '-', etc). By doing this, we actually use symbols to replace original relationships which connect objects together.  For simplicity, we can pre-defined the set of symbols, and the mapping between each kinds of symbols to each kinds of relationships.
>
> What's more, although we only verified on specific task examples, the idea itself can provide insights for future research: (e.g. customizing the conversion process, customizing mapping between symbols and relationships, even customizing the way of using symbols in in-context learning), what important is, we provide insights to **use symbols to replace natural languages** in demonstration / prompting, which **would be a promising direction to further explore for future works**, for using symbols can wider the representation space in prompting, and reducing potential noises brought from natural languages.
>
> Point 2:
>
> Based on guideline we give in the point 1 and in the section 3 of the paper, we can easily transform any sequential natural language exemplars which meet the conditions to chain-of-symbol exemplars. The guideline is suitable for all tasks which meet the conditions, we have proved its effectiveness in spatial relationships and temporal sequence reasoning tasks as examples, and future work can explore more tasks.
>
> Point 3:
>
> For open-source models, the results and analyses are concluded in part 1 of general response for all reviewers. Figure 4 in our latest uploaded version of paper gives a clear results about scaling curves of CoS compared with CoT, it shows CoS has more steep curve for all tasks, which indicates it is a general ability that LLMs can understand abstract symbols better when size increases.
>
> **For questions**:
>
> Point 1:
>
> We conclude this point in the second part of general response for all reviewers, specifically, we decompose this into two dimension: task domain and task complexity.
>
> For task domain, inspired from Reviewer c5jB, we use similar scenarios in BBH benchmark, called "temporal sequence", it is a reasoning task which provided a sequence of time slots and related scenarios, and model should answer which time is available based on given descriptions. And in the above second table, we can see that using CoS can outperform CoT significantly (from 70 to 93.) We also conduct experiments on GPT-3.5-Turbo, and shows CoS can outperform CoT from 66.5 to 76.2
>
> For tasks complexity:
>
> We conduct control experiments where we control the complexity of the task by setting the number of bricks in brick-1d scenarios. When the number of bricks increases, which means the complexity of the task increases, the performance of CoS can outperform CoT. It verify the robustness of CoS when task becoming hard.
>
> For whether there is an automatical way to convert or not, see the point 1 for weakness above.
>
> Point 2:
>
> For other methods, for example program-based CoT, can you be more specific? We only find this related work https://openreview.net/pdf?id=YfZ4ZPt8zd, which uses an external program executor to help calculate, but to be honest it is not a fair comparison with our CoS method which only does the inference by LLMs on their own, but we can list these kinds of works in our related work section.
>
> Point 3:
>
> The computation required to convert symbols is down to 10ms through Python program based on given CoT exemplars. For those tasks which do not have manual annotated symbols, we can follow the procedure described in the part 1 of weakness above. For a broader range of tasks, just as the description in part 2 of general response for all reviewers, we explore temporal sequence tasks and verified the effectiveness of our method.

---

> > ### Comment · Reviewer_tveC · 2023-12-04
> > **Thanks for the detailed response.**
> >
> > Thanks for the detailed response. Most of the concerns are addressed, and in general I do favor this paper idea. I raise from 5 to 6.

---

> ### Author Response · Authors · 2023-11-23
>
> Dear reviewer tveC,
>
> We greatly appreciate your valuable feedback. We conduct additional experiments and write a detailed shared response and specific response for your review.
>
> As you may know, unlike previous years, the discussion period this year can only last until today (only 3 hours left).
>
> If you are satisfied with our response, please consider updating your score. If you need any clarification, please feel free to discuss with us.
>
> Best,
>
> Authors

---

### Official Review · Reviewer_Doc8 · 2023-11-05

**Soundness:** 2 fair
**Presentation:** 3 good
**Contribution:** 2 fair
**Rating:** 6
**Confidence:** 4

**Summary:**

The authors first investigated the performance of ChatGPT on spatial relationship tasks. They find that it still struggles, and proposed a novel prompting method called Chain-of-Symbols (CoS), which converts Chain-of-Thought (CoT) prompts into a sequence of symbols that represent spatial relationship described in the problem of interest. They experimented with synthetic datasets (Brick World, NLVR-based Manipulation, and Natural Language Navigation) and more realistic dataset (SPARTUN), and showed that CoS improves over CoT.

**Strengths:**

- Show that LLMs still struggle with spatial relationship understanding from natural language
- Proposed a new prompting method called Chain-of-Symbol to improve the performance on spatial relationship tasks.

**Weaknesses:**

- The paper focuses only on Text-Davinci-003 and GPT-3.5-turbo. However, it would be beneficial to include other promising open-source large language models like the Llama-2 series in their study to see the effect of model size, different base models, etc.
- The authors claim that they achieved a significant improvement of up to 60.8% in accuracy (from 31.8% to 92.6%) on the Brick World dataset. While this is technically correct given the results, the reported improvement is a little misleading because it's comparing zs-CoT vs. CoS and also such a dramatic gain isn't seen in most cases, especially when comparing the performance of CoS to CoT. In more realistic datasets like SPARTUN, the improvement is marginal. This suggests that the impressive gains are more about the simplicity of the Brick World dataset than any substantial leap in CoS performance. Thus, I believe the claim should be more moderate.
- Apart from the previously mentioned issues, I find the contribution is limited for the following reason: the paper does demonstrate how you can condense the chain of thought (CoT) into a series of symbols, and that this is enough for tasks involving spatial relationships. However, this method of prompting might only be effective for spatial relationship tasks. Also, when you look at the final experiment (Table 5), it suggests that this method might not work as well in real-life situations. It seems that the success of this method, referred to as CoS, might be limited to simpler, more straightforward spatial tasks.

**Questions:**

- The authors used GPT-4 exclusively for the final experiment. Why not present GPT-4 results in the other sections as well?

---

> ### Author Response · Authors · 2023-11-15
>
> Thanks for your reply and advice, we follow your advice and complete several new experiments, which described in the general response for all reviewers. Below is some response for weakness and question.
>
> **For weakness 1**:
>
> You can see the part 1 of general response for all reviewers, we add the experiments of open-source LLMs, llama-2 with different sizes. And you can also see the figure 4 in latest uploaded version of our paper very clearly about the more steep scaling curve of our method across different tasks compared with CoT. It shows the effectiveness of our method compared with CoT and the enmergent ability of understanding CoS.
>
> **For weakness 2**:
>
> Thanks for pointing out our misleading part, We revised our claim in the revision version.
>
> For the concern of marginal improvements, we add further experiments about using longer description instances (e.g. from 6 to 14 bricks) and find that CoS can gain more improvements compared with CoT in these scenarios compared with more simple scenarios, which is reported in part 2 of general response for all reviewers.
>
> For "the impressive gains are more about the simplicity of the Brick World dataset", actually the "shuffle both" results of brick 1d shows it is not a simple task as it looks like, where CoT only gains 43.0 accuracy. This actually shows that LLMs do not have good spatial understanding capabilities, because they cannot do spatial reasoning tasks which human think is easy well.
>
> **For weakness 3**:
>
> We add experiments on the temporal sequence tasks from the BBH benchmark [1]. This is another task which is not related with spatial relationships, and which also contains a more real-life situation. We report the results in table 2, part 1 of general response for all reviewers, and also tested on GPT-3.5-Turbo, and shows CoS can outperform CoT from 66.5 to 76.2.
>
> "It seems that the success of this method, referred to as CoS, might be limited to simpler, more straightforward spatial tasks."
> We use complexity control experiment and task of temporal sequence of BBH benchmark to shows the contribution of our proposed method, and we also want to describe our method in a more general way:
>
> We will add this into Section 3 in our paper: we can formulate the tasks which our method is suitable as a set of tasks which has sequential and limited number of relationships, and the relationships can be spatial, and also can be other kinds such as temporal.
>
> For converting natural languages into symbols, we can use human, NER system or even rule-based codes to automatically recognise objects in the sequential relationships described by natural languages, and then connect them with each other using random symbols ( e.g. '/', '-', etc). By doing this, we actually use symbols to replace original relationships which connect objects together.
>
> Generally, we want to provide insights about using symbols in demonstration of the prompting, and we provide a particular way to use symbols to make LLMs to be more effective and efficient when handle sequential tasks.
>
> **For question:**
>
> We only have limited access of GPT-4 API currently, we will add the related results when we can access GPT-4 API in our future version.
>
> [1] Challenging BIG-Bench Tasks and Whether Chain-of-Thought Can Solve Them. ACL 2023 Findings

---

> > ### Comment · Reviewer_Doc8 · 2023-11-23
> > **Reply**
> >
> > Thank you for your responses and additional experiments. My questions regarding generalizability of this method and other LLMs have been addressed.
> > The scaling curve that shows CoS improves more on larger models (e.g. 70B or more) across different datasets (Brick 1D, Navigation, and Temporal Sequence) is indeed interesting.
> > Regarding GPT-4 API, it has been accessible to everyone at least for the past few months. I've updated my score assuming that GPT-4 experiments will be included in the final revision.

---

### Author Response · Authors · 2023-11-15
**General response for all reviewers**

We first thank for all reviewers to provide valuable advices, which encourages us to have more findings.

We are glad that reviewers think our work is simple but effective, and novel: Reviewer tveC said "a novel method", Reviewer c5jB said "The proposed method is very simple, but the results show a significant improvement", Reviewer LjHX said "findings are very interesting!".

We are also glad that reviewers think our work is easy to read and understand: Reviewer c5jB said "easy to understand. All the sections are easy to read", Reviewer LjHX said "well-organized and easy to read."

Then we aim to solve **major common concerns** provided by reviewers, and we concluded into three main parts:

1. Results of open-source models: provided by Reviewer Doc8 and Reviewer tveC

2. Scalability and Generalization of our proposed method: provided by **all reviewers**

3. Fundamental explaination and insights: provided by Reviewer c5jB and Reviewer LjHX

We solve them one by one as below:

**1. Results of open-source models:**

We use llama-2 as our representative open-source LLM, we test our method in all sizes (7B, 13B, and 70B) in the three tasks for limited time and computing resource.

Brick-1D:
| Method / Model Size | 7B | 13B | 70B |
| ------- | ------- | ------- | ------- |
|   CoT  |   0.04  |  0.21    |  0.50    |
|  CoS    |  0.02    |  0.09    |  0.69    |

Navigation:
| Method / Model Size | 7B | 13B | 70B |
| ------- | ------- | ------- | ------- |
|   CoT  |   0.17  |  0.23    |  0.45    |
|  CoS    |  0.12    |  0.18    |  0.53    |

Temporal Sequence:
| Method / Model Size | 7B | 13B | 70B |
| ------- | ------- | ------- | ------- |
|   CoT  |   0.15  |  0.33    |  0.70    |
|  CoS    |  0.20    |  0.29    |  0.93    |

Temporal-sequence is the new tasks we inpired from the review of Reviewer c5jB, we find similar scenarios in BBH benchmark [1], and tested on it to further verify the effectiveness and generalization of our method.

In these three tables, we surprisingly find that although in 7B and 13B, our method is lower than or near to the performance of CoT, our method can outperform CoT in 70B clearly in all tasks!  This indicates:

1. The **scaling curve** of CoS is steeper than CoT’s.

2. It shows the **emergent ability** of LLMs for understanding CoS (symbols) when size increases.

We also upload the figure about scaling curve in the **figure 4** in the latest version of our paper, which is very clear and impressive.

The experimental results matched with the intuition that larger models have stronger basic capabilities, so they might have the ability to understand complex parterns represented by symbols.

**2. Scalability and Generalization of our proposed method:**

We decompose this into two dimension:

task domain and task complexity

For task domain:

As described above, we inpired from the review of Reviewer c5jB, use similar scenarios in BBH benchmark, called "temporal sequence", it is a reasoning task which provided a sequence of time slots and related scenarios, and model should answer which time is available based on given descriptions. And in the above second table, we can see that using CoS can outperform CoT significantly (from 70 to 93.) We also conduct experiments on GPT-3.5-Turbo, and shows CoS can outperform CoT from 66.5 to 76.2

For tasks complexity:

We conduct control experiments where we control the complexity of the task by setting the number of bricks in brick-1d scenarios. When the number of bricks increases, which means the complexity of the task increases, the performance of CoS can outperform CoT. It verify the robustness of CoS when task becoming hard.

Complexity control experiment on GPT-3.5-Turbo:
| Method / Length| 6 | 8 | 10 | 12 | 14 |
| ------- | ------- | ------- | ------- | ------- | ------- |
|  CoT    |  0.62    |  0.56    |  0.40    |  0.35   |  0.32    |
|   CoS  |   0.64  |  0.62    |  0.56    | 0.52    | 0.50    |

**3. Explanation and insights:**

We use CoS for two reasons:

1. Using symbols rather than text words can improve the representation space in prompting.

2. CoS uses symbols to replace parts of natural languages, hence reducing noises brought from redundant natural languages.  One assumption is that words of natural languages might have more context information during pre-training, which might bring potential additional information, but using symbols can avoid these, by connecting objects without potential additional information, and make prompting more clear and structural.


[1] Challenging BIG-Bench Tasks and Whether Chain-of-Thought Can Solve Them.  ACL 2023 Findings

---

### Comment · Area_Chair_Ya4o · 2023-11-16
**Read Authors' Responses and Reply Accordingly**

Hi,

Thanks for your help reviewing this paper.

The authors have provided a detailed response to each of your reviews as well as a shared response to all reviewers. Please be sure to read through their responses and respond accordingly so that we can have a healthy discussion of the merits of this paper.

Best,\
AC

---

> ### Comment · Area_Chair_Ya4o · 2023-11-20
> **Bump...**
>
> There are only **two days remaining** for the author-reviewer discussion (November 22nd). Please read through the authors' response to your review and comment on the extent to which it addresses your questions/concerns.

---

### Author Response · Authors · 2023-11-23

Dear reviewers,

There are only several hours left. Thank you for your review. We added experiment results and responses to your review several days ago. And we really hope you can reply our response or discuss with us. Thanks.

Best,

Authors

---

### Meta-Review · Area_Chair_Ya4o · 2023-12-13

**Metareview:**

The paper considers the ability of large language models (LLMs) to perform tasks that involve spatial reasoning in support of planning. The paper highlights deficiencies with existing methods and proposes Chain-of-Symbols (CoS), a framework that converts Chain-of-Thought (CoT) reasoning into a sequence of symbols that capture relevant spatial relationships. The method is evaluated on a series of synthetic datasets and shown to outperform a CoT baseline.

The paper is highly topical---much attention is being paid to the ability to utilize LLMs for robot reasoning and a better understanding of their spatial reasoning capabilities is of interest to the community. Beyond the paper's relevance, the reviewers appreciate the simplicity of the method and its improvements over the CoT baseline, as well as the clarity of the presentation. However, the performance gains primarily result from the approach to prompt engineering and the algorithmic contributions of the work are unclear. The authors acknowledge this in their response, which the AC appreciates, yet the suggested benefits to future work are not obvious.

**Justification For Why Not Higher Score:**

The paper does not contribute new algorithms nor are the insights into how LLMs can be used for spatial reasoning particularly impactful.

**Justification For Why Not Lower Score:**

N/A

---

### Decision · Program_Chairs · 2024-01-16

Reject